# An Intelligent and Precise Agriculture Model in Sustainable Cities Based on Visualized Symptoms

Bashar Igried [1], Shadi AlZu'bi [2], Darah Aqel [2], Ala Mughaid [1], Iyad Ghaith [2] and Laith Abualigah [3,4,5,*]

1  Department of Information Technology, Faculty of Prince Al-Hussien Bin Abdullah II for IT, The Hashemite University, Zarqa 13133, Jordan
2  Faculty of Science and IT, Al-Zaytoonah University of Jordan, Amman 11733, Jordan
3  Computer Science Department, Prince Hussein Bin Abdullah Faculty for Information Technology, Al al-Bayt University, Mafraq 25113, Jordan
4  Hourani Center for Applied Scientific Research, Al-Ahliyya Amman University, Amman 19328, Jordan
5  MEU Research Unit, Middle East University, Amman 11831, Jordan
*  Correspondence: aligah.2020@gmail.com

**Abstract:** Plant diseases represent one of the critical issues which lead to a major decrease in the quantity and quality of crops. Therefore, the early detection of plant diseases can avoid any losses or damage to these crops. This paper presents an image processing and a deep learning-based automatic approach that classifies the diseases that strike the apple leaves. The proposed system has been tested using over 18,000 images from the Apple Diseases Dataset by PlantVillage, including images of healthy and affected apple leaves. We applied the VGG-16 architecture to a pre-trained unlabeled dataset of plant leave images. Then, we used some other deep learning pre-trained architectures, including Inception-V3, ResNet-50, and VGG-19, to solve the visualization-related problems in computer vision, including object classification. These networks can train the images dataset and compare the achieved results, including accuracy and error rate between those architectures. The preliminary results demonstrate the effectiveness of the proposed Inception V3 and VGG-16 approaches. The obtained results demonstrate that Inception V3 achieves an accuracy of 92.42% with an error rate of 0.3037%, while the VGG-16 network achieves an accuracy of 91.53% with an error rate of 0.4785%. The experiments show that these two deep learning networks can achieve satisfying results under various conditions, including lighting, background scene, camera resolution, size, viewpoint, and scene direction.

**Keywords:** agriculture intelligence; precise agriculture; sustainability; predefined model; computer vision; intelligent systems





## 1. Introduction

Harvest diseases are the main reason for starvation and food instability in many countries. It is estimated that the annual losses or damages in the crop yields due to bacteria, viruses, and microorganisms is up to 16% globally [1]. Moreover, the solutions currently used to combat various plant diseases require massive use of yield protection products, which expose the environment and users to different threats. The traditional methods, which use a microscope and DNA, are still significant in identifying and discovering different diseases. Even though many farmers worldwide do not consult traditional diagnostics tools, most have a smartphone. Telco companies forecast that mobile subscribers will hit 8.94 billion in 2027, while 86% of the subscriptions will be smartphone subscriptions [2]. Therefore, a software tool based on a smartphone device is a candidate solution, which helps diagnose yield diseases depending on a plant leaf's captured and analyzed picture.

In general, many previous researchers have employed artificial intelligence (AI) in real-life applications, including agriculture, business intelligence, transportation, health-care, social media, smart cities, and many more [3–8]. Overall, our study and discus-

sion on AI-based deep learning in the right direction can be a reference guide for future research and development in the plant disease applications domain by academics and industry professionals.

The main objective of using machine learning (ML) in precise agriculture to predict leaf disease in sustainable cities is to improve crop yields and reduce losses caused by disease. By using ML algorithms to analyze large amounts of data and identify patterns, farmers can predict the onset of disease in crops, allowing them to take preventative measures before the disease becomes severe. Additionally, ML-based systems can improve disease diagnosis efficiency by automating the process, reducing the time and labor required to inspect crops manually for signs of disease. Furthermore, by using ML in precision agriculture, farmers can optimize resources such as water and fertilizers, reducing the environmental impact of farming practices. Increasing crop yields can also help reduce food insecurity in sustainable cities. In our study, we focus on classifying apple leave diseases. Apple trees are threatened by diseases such as apple scabs, sooty blotch, flyspeck, cedar apple rust, and powdery mildew. Apple scab is caused by a fungus that leads to infected leaves. The responsible bacteria usually thrive in cold, moist climates. While the sooty blotch and flyspeck may shorten the apple's shelf life [9].

Due to the limited resources and expertise in the science of the causes and effects of diseases worldwide, there is a necessity for a system that can automatically identify and classify apple diseases in batches. In addition, the advance in the Internet of Things (IoT) worldwide and the widespread 5G technology make it easily accessible to farmers to take immediate action if disease symptoms appear.

Many computer vision and machine learning techniques have been employed to detect diseases that can hit certain crops, such as tomatoes and potatoes. These techniques are motivated by advancements in Convolutional Neural Networks (CNN) and computer vision in general. Since CNN can produce superb results in classifying images, the proposed says disease diagnosis efficiency CNN for plant disease identification and classification.

In this paper, over 18,000 images from the Apple Diseases Dataset by PlantVillage have been tested, including healthy and affected apple leaves. the main contributions of this work are as follows :

- The efficient use of the convolutional neural network (i.e., VGG-16 architecture) to unlabeled plant leave images dataset.
- Employ pre-trained deep learning techniques, including Inception-V3, ResNet-50, and VGG-19, to solve visualization problems in computer vision, including object classification.
- Training the dataset with these pre-trained models and comparing the accuracy and error rate between them.
- The efficient use of preprocessing stages such as histogram equalization to achieve more promising results.

Inspecting crops for signs of disease or model requires the most miniature image preprocessing overhead. Our model different feature maps directly from images. The experimental model can distinguish between four types of apple leaves. The remaining parts of the paper are organized in the following, or Section 2; the related work is presented. Section 3 explains the methodology, and the convolutional neural network mode experimental evaluation and analysis are reported in Section 4. Finally, Section 5 presents the paper's conclusion and future work.

## 2. Literature Review

Numerous academics have suggested using feature maps and image processing as identification tools in agriculture applications to spot dangerous grass in fields and identify different illnesses [10,11]. Research on the automated detection of plant diseases is frequently discussed because it might provide the expected advantages of monitoring broad areas of crops and quickly identifying the visual indications of illnesses on trees and their leaves.

Therefore, searching for a reliable, affordable, and accurate way to diagnose apple's illness is smart. Using excessive chemicals to cure plant diseases would boost operating expenses and increase the risk of toxic levels that harm crops. This answer calls for accurate illness identification and mention of the disease's stage. Consequently, a reliable illness identification system is required.

The authors of paper [12] suggested a K-means segmentation algorithm-based image processing technique for the early identification and categorization of five different leaf diseases. They employed a statistical classification and the back-propagation technique to categorize the studied disorders. Additionally, they used the color co-occurrence approach to take both texture and color characteristics into account while extracting feature data.

The authors of [13] described a technique for transforming the RGB picture of the diseased plant leaf into the color transformations H, I3a, and I3b. The transformed image was then segmented by looking at the intensity distribution in a histogram. The extracted region underwent further processing to eliminate pixel portions not considered part of the region of interest. The gradient of the adjacent pixels was then examined. Finally, the suggested technique marked the unhealthy areas.

The authors of [14] provided a method for identifying illnesses in the leaves of grape plants. They started using the K-means clustering technique to grade diseases and segment images. The Gray Level Co-occurrence Matrix (GLCM) was also employed to extract the textural characteristics from the retrieved illness part. Then, the Spatial Gray-level Dependence Matrix (SGDM) was utilized to forecast if a pixel at one particular gray level will appear at a different distance and orientation from any given pixel. The feed-forward back-propagation neural network approach was then used to categorize the leaf diseases affecting grape plants.

The authors reported a technique for the automated identification and categorization of plant leaf diseases in article [15]. They began by identifying the pixels that were primarily green in color. Then, using Otsu's approach to estimate the threshold values, these pixels were muted depending on those values. In order to create a distinctive feature set that accurately depicts a particular image, they also employed the Color Co-occurrence Method (CCM) to extract features based on an image's color and texture attributes.

Using the extreme learning machine (ELM) algorithm on a collection of plant leaf photos, Aqel et al. [16] demonstrated a method for identifying plant leaf diseases. In addition, they employed the GLCM for feature extraction and the k-means clustering technique for picture segmentation. The study also used the BDA optimization technique for feature selection. The tests revealed that their method successfully classified plant leaf diseases with an efficiency of 94%, yielding positive results.

The authors in [17] used the multilayer perceptron (MLP) neural network and a hybrid metaheuristic feature selection method for detecting mango leaf diseases. In contrast, the authors in [18] described a technique that applied ELM with the simulated annealing optimization (SAO) algorithm to classify plant leaf diseases in Jatropha Curcas. The applied technique achieved encouraging results.

The authors gave a method for detecting plant diseases using deep ensemble neural networks in article [19]. Their method fine-tuned multiple pre-trained neural network models using the classification technique, including ResNet-50, InceptionV3, DenseNet, MobileNetV3, and NasNet. The performance of DENN beat the other pre-trained neural network models when their methodology was applied to the Plant Village database.

The authors disclosed a method for identifying and categorizing potato plant leaf disease in publication [20]. They used the K-means clustering technique for image segmentation. For feature extraction, they used the GLCM. The Plant Village dataset was used to apply the support vector machine (SVM) method, producing a classification accuracy of 99.99%.

The authors gave a model for identifying plant leaf diseases by employing an ideal mobile network-based convolutional neural network in publication [21]. The classification process employed bilateral filtering (BF) in their investigation, while the picture

segmentation stage used Kapur's thresholding. Moreover, the MobileNet model was implemented as an extraction of features approach. The emperor penguin optimizer (EPO) algorithm improved the hyperparameters and boosted the crop disease classification rate. Lastly, the ELM classifier was employed for allocating the proper classification to the plant leaf pictures. The OMNCNN model, according to the authors, had a high accuracy of 98%.

Leaf disease prediction in smart and precise agriculture has been a growing interest in recent years, as it offers the potential to increase crop yields and reduce losses caused by disease [22,23]. However, despite the advancements in technology and machine learning (ML) techniques, there remains a gap in the literature effectiveness of these systems in predicting leaf diseases. One major gap in the literature is the lack of robust and reliable ML models for leaf disease prediction. While many studies have proposed various ML-based approaches, most have been tested on a limited number of disease classes and small datasets, making it difficult to generalize their performance to other diseases or crops. Additionally, many proposed methods lack thorough performance evaluation, making it challenging to compare their effectiveness with other approaches [24,25].

Another gap in the literature is the limited attention given to the practical considerations of implementing leaf disease prediction systems in smart agriculture. While the development of ML models for disease prediction is important, there is a lack of research on the technical, economic, and social factors crucial for the successful deployment of these systems in real-world settings. This includes data privacy and security, scalability, and the cost and feasibility of large-scale deployment of these systems. Furthermore, there is a shortage of studies that assess the impact of leaf disease prediction systems on overall agricultural production and farmers' livelihood. Most existing studies focus on the technical aspects of the models, with little attention given to the potential benefits or drawbacks these systems can have on the agricultural sector and the farmers that would be using them [26].

AI has been employed efficiently in many agricultural applications [27–29]. In conclusion, while leaf disease prediction in smart and precise agriculture holds great potential, there is still a significant gap in the literature on the effectiveness of these systems. To fully realize the potential of these systems, research is needed to develop robust and reliable ML models, consider practical implementation considerations, and evaluate the impact on agricultural production and farmers' livelihood.

This research paper presents a model for sustainable agriculture in cities. The paper proposes a novel approach that combines visual symptom recognition and machine learning techniques to accurately diagnose plant diseases and pests in urban agriculture. The paper outlines the limitations of traditional agriculture models and the need for a more sustainable and intelligent approach to agriculture in cities. The authors propose an intelligent agriculture model that utilizes visual symptom recognition to diagnose plant diseases and pests. The model is based on a deep learning algorithm that can accurately recognize and classify visual symptoms of plant diseases and pests. The authors also present a case study where the proposed model was used to diagnose tomato plant diseases in an urban agriculture setting. The results of the study showed that the proposed model was able to accurately diagnose plant diseases with a high level of precision. Overall, the paper presents an innovative approach to sustainable agriculture in cities. The proposed model has the potential to revolutionize urban agriculture by providing an accurate and intelligent solution for diagnosing plant diseases and pests.

## 3. Methodology

The proposed system aims to classify a disease category in apple trees' leaves. Primarily, it aims to classify whether a leaf belongs to a healthy, multiple-diseased, scab, or rust category, using image processing and a deep learning (DL) pre-trained model.

The general methodology for using ML in plant leaves disease prediction typically involves several steps:

1. Data collection: Collect a large dataset of images of healthy and diseased leaves, along with labels indicating the present disease type. This Data can be collected from various sources, such as field observations, experiments, or online databases.
2. Data preprocessing: Clean and preprocess the collected data to ensure it is suitable for training the ML model. This may involve resizing images, removing noise, or balancing the dataset to ensure equal samples for each class.
3. Feature extraction: Extract features from the images relevant to the disease prediction task. This may involve using edge detection, color histograms, or texture analysis.
4. Model selection and training: Select an appropriate ML algorithm and train it on the preprocessed data. This may involve using techniques such as supervised learning, unsupervised learning, or deep learning.
5. Model evaluation: Evaluate the performance of the trained model on a hold-out dataset. This may involve using accuracy, precision, recall, or F1 score metrics.
6. Model deployment: Once the model has been trained and evaluated, it can be deployed to predict new images of leaves.
7. Model maintenance and updating: Continuously monitor the model performance and update it with new data and the latest techniques to improve its performance over time.

It is essential to note that this is a general methodology, and the specific steps may vary depending on the type of data and the specific disease prediction task. It is also important to validate the predictions with ground truth data before deploying the model.

To work with the mentioned issue, a deep learning-based methodology is introduced to specify whether apple leaves have diseases. The system design is illustrated in Figure 1, which displays that the model includes two key parts: image processing and a DL classification model of apple leaves.

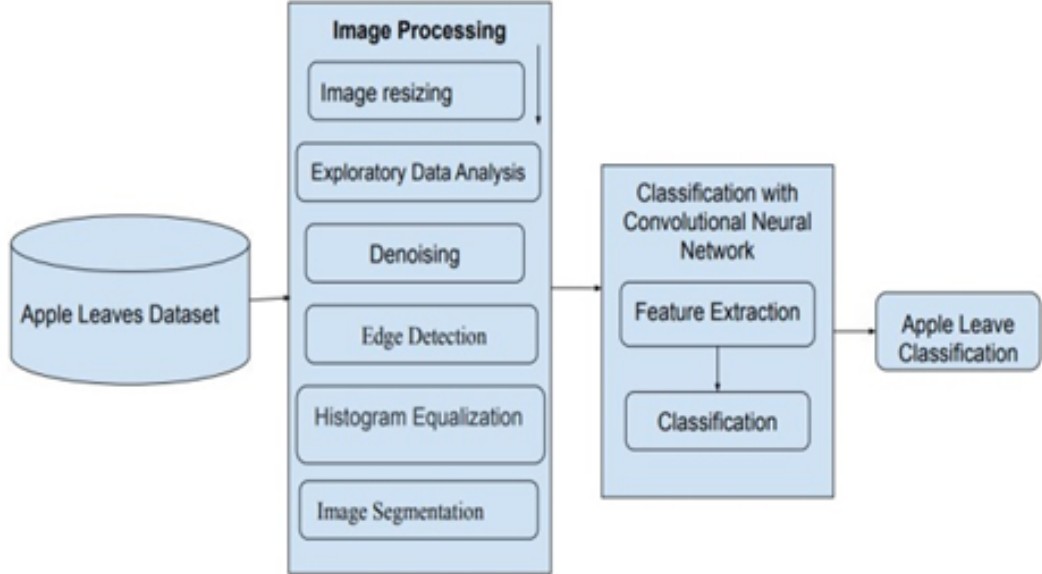

**Figure 1.** The Proposed Framework.

*3.1. Dataset*

The proposed system has been tested using the most famous apple leaf disease dataset, which is the "Apple Diseases Dataset" by PlantVillage, a research and development project led by Penn State University in collaboration with the AI company Cognitivescale [30]. This dataset includes images of healthy apple leaves as well as those affected by various diseases, such as apple rust, apple scab, and powdery mildew. The dataset contains over 18,000 images and has been used for training and testing machine-learning models in plant disease diagnosis and classification. The Apple Diseases Dataset is widely recognized as a valuable resource for researchers and practitioners in agriculture and machine learning.

Apple orchards face many threat agents, such as insects and pathogens. Therefore, timely and suitable pesticide spraying depends on detecting disease early. False and late recognized symptoms detection may lead to either extra or little usage of chemical materials. Images dataset was captured by Ranjita Thapa et al. in [31] for 3651 high-quality, real-life symptom images of multiple apple foliar diseases with different lighting levels, viewpoints, and backgrounds.

Each training image belongs to one of the following four labels: Healthy, Multiple-diseased, Rust, or Scab, as shown in Figure 2. While the testing images are captured from the same source distribution, the aim is to label those testing images with one of the four labels.

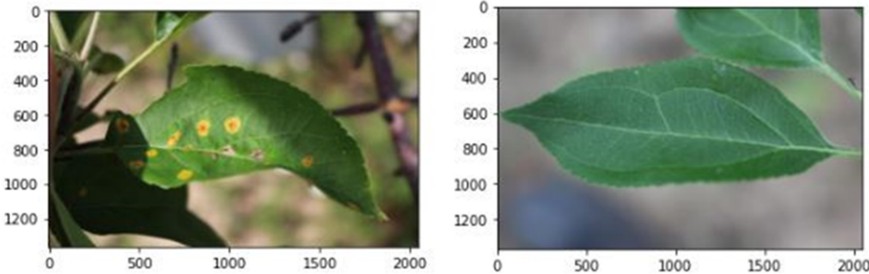

**Figure 2.** Leaves examples.

### 3.2. Image Enhancement

The image is enhanced by five iterations of the following image filters.

### 3.2.1. Denoising Filter

Image smoothing methods such as Gaussian and Median Blurring are used in removing noise to an acceptable degree. In such methods, neighbors around a pixel are taken and filtered using the Gaussian smoothing filter, which can preserve the edges better than the same-sized mean filter. The other filter used in removing noise is the median filter of the values etc., to replace the value of the neighboring values with the median value. Figure 3 illustrates an image before applying the Gaussian filter and the resulting de-noised image.

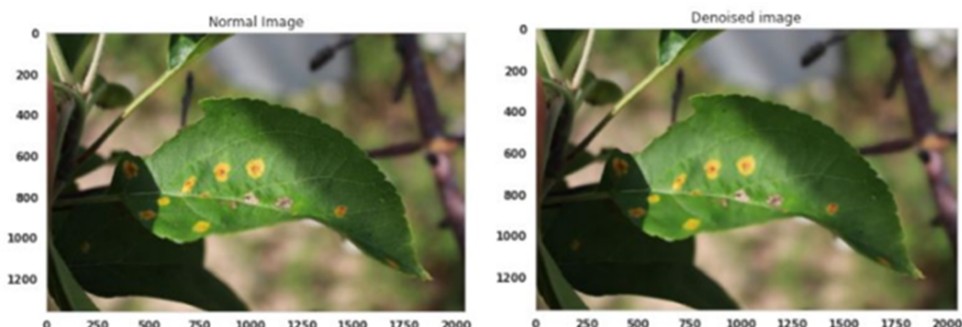

**Figure 3.** Applying the De-noising Filter on Images.

### 3.2.2. Edge Detection

One of the main operations of image processing is edge detection. Edge detection relies on locating the sharp cuts or gaps in images, such as in-depth or surface orientation or changes in material properties and differences in illumination in the scene. When applying it, the amount of pixels is decreased while keeping the image's most important features. The main function of edge detection is the convolution operation. Figure 4 illustrates how to manipulate the convolution operation. Edge detection is performed using the Sobel filter function, tested on the sample images as illustrated in Figure 5.

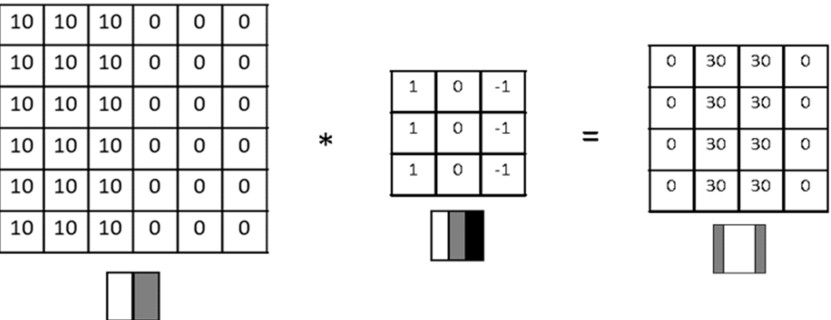

**Figure 4.** Edge Detection Using Convolution.

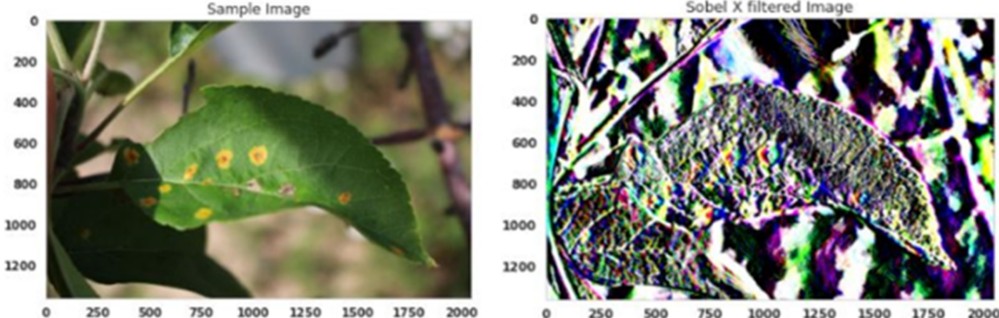

**Figure 5.** The Sobel Filtering Method.

### 3.2.3. Multi-Stage Edge Detectiong Cannay Filter

The Canny filter depends on a Gaussian filter's first and second derivatives to calculate the intensity change.

The Gaussian low-pass filter is used to reduce the effect of noise present in the image as a first step in detection to smooth the image by removing noise. In the second step, the potential edges are sharpened based on the direction edges and the gradient magnitude by applying the Sobel filter at a pixel level. Finally, the double threshold is applied to separate strong, weak, and suppressed edges. Weak edges are preserved or suppressed using edge tracking. Figure 6 shows the image after applying the multi-stage edge detection filter.

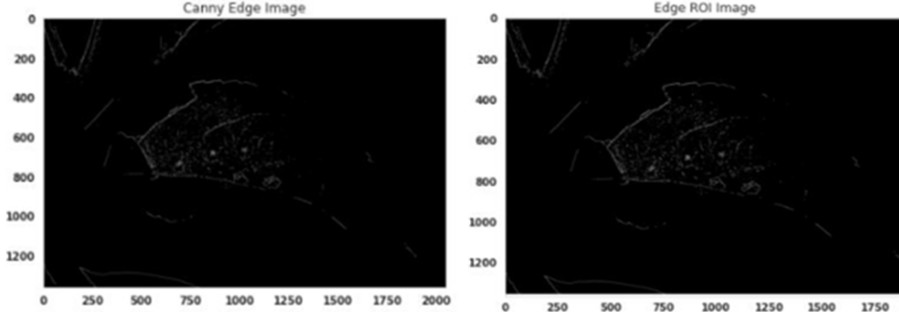

**Figure 6.** Canny Edge Image, Edge ROI Image.

### 3.2.4. Local Histogram Equalization

Instead of the RGB color bits, equalization provides the intensity levels of a picture. The channels of a detailed RGB color picture cannot simply apply the histogram. The filter must be applied to the equalized pixel intensities without changing the RGB color values, as shown in Figure 7.

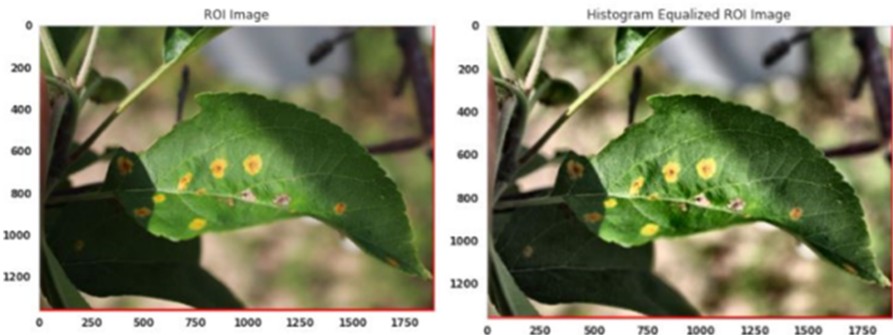

**Figure 7.** Image Enhancement Using the Local Histogram Equalization on ROI Image.

### 3.2.5. Image Segmentation (Colored Images)

K-means clustering is used to segment different areas of the image based on centroids. The clustering is done on the histogram-equalized image of an apple leaf. Figure 8 illustrates the final image segmentation using the K-means technique, which is applied to the histogram image. K-means is considered helpful for applications such as image compression and object realization because, for these applications, it is unnecessary to manipulate the entire image.

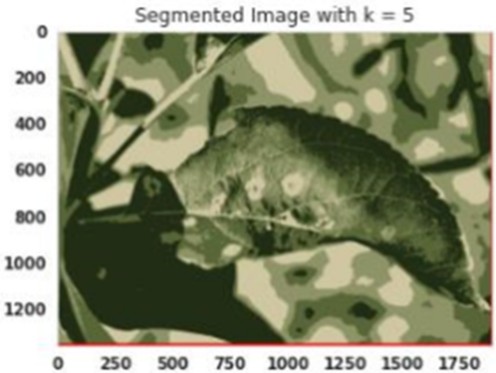

**Figure 8.** Image Segmentation Using K-means Clustering.

### *3.3. Transfer Learning and DL Model Tuning*

Transfer learning is mainly used to train a deep learning network through less Data than it may need if it must be trained from the beginning. Using transfer learning, the knowledge is transferred from a model which has been learned previously to the proposed model. While transfer learning, as shown in Figure 9, has been proven to minimize the required time for training the dataset, many convolution networks, such as VGG-16 neural network architecture, are being used to learn the model.

### 3.3.1. Model Architecture

The ImageNet dataset and its VGG-16 model has originally 16 layers of convolution, max pooling, and fully connected layers. Figure 9 shows the VGG-16 different layers which compose the pre-trained model. The ImageNet database also contains images of the fixed size (224 × 224 × 3) to represent the RGB channels. Therefore, the tensor of (224, 224, 3) is the input. This model processes the input images and outputs a vector of 1000 values.

Convolution Neural Network consists of three key parts: the convolution layers, the max-pooling layers, and the fully connected layers, as shown in Figure 10. The convolution and max pooling layers extract the feature set from the given images, while the fully connected layer classifies the labels.

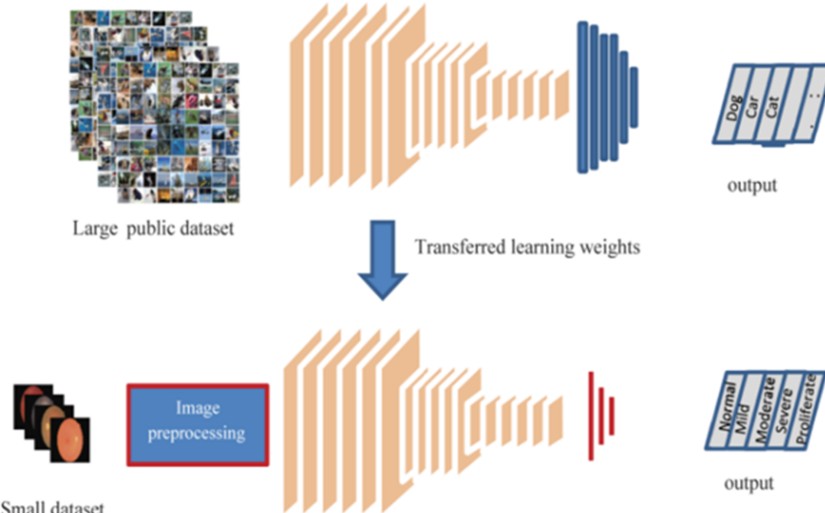

**Figure 9.** Transfer Learning Architecture from [32].

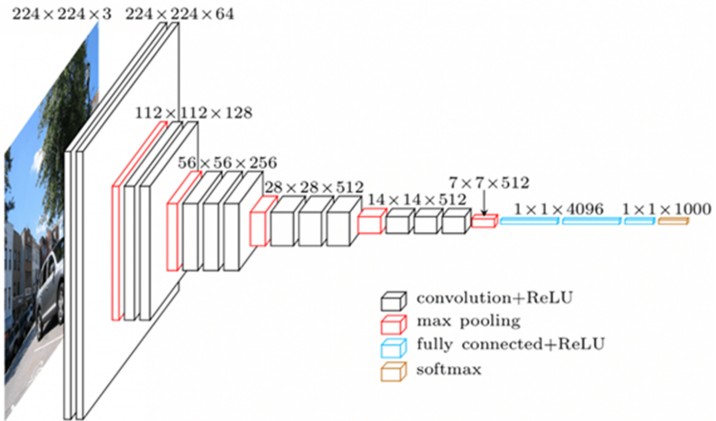

**Figure 10.** VGG-16 Network Model from [33].

### 3.3.2. Convolutional Neural Network (CNN)

CNN is a deep learning architecture capable of analyzing input images by assigning weights and biases to distinctive features, enabling it to differentiate them from other features. CNN network has the following layers:

- Input layer: this layer has nothing to learn.
- Convolution layer: this is where CNN learns.
- Max Pooling layer: this layer has neither weights nor trainable parameters.
- Fully Connected Layer (FC): this layer has trainable parameters. This layer has the highest number of parameters.

### 3.3.3. Feature Extraction

Using features such as Healthy, Multiple Diseased, Rust, or Scab can be very useful in plant disease classification because they are distinct visual symptoms that can aid in identifying the specific disease affecting a plant. The Healthy feature is used as a baseline comparison for the other features. By examining a healthy plant's characteristics, such as its leaf color, size, and shape. Multiple Diseased features are useful when a plant has more than one disease affecting it simultaneously. The presence of multiple diseases can create unique visual symptoms that may not be observed when only one disease is present. Extracting a high-level feature set from the input images is an important task performed by a network. The model in Figure 11 illustrates a set of convolution and pooling layers connected in series to perform this task.

- The Convolution Map

The purpose of the convolution map is to extract a feature set from a given image. This step has a group of trainable kernels, where each kernel is implemented at the pixel level of the image for each raw. The RGB color channels in the sliding window are considered by computing the dot products between the kernel pixel and the input image pixel. The output result of the activation map of the kernel is called the feature map. Then, the CNN model will learn features such as edges, curves, and texture that will trigger when the network finds the known features in the input image. During the training stage, the network will learn the values of these features by itself.

- Max-pooling Map

A sub-sampling layer comes after the convolution layer. This layer will reduce the size of the generated convolution feature set. The output of the max-pooling layer is provided by an activation function in the input layer over sub-windows within each feature map. It is used to minimize the size of a feature map.

- The Classification Step

The last layers in the CNN network are called the fully connected layers. These layers interconnect each neuron to all extracted feature sets in the preceding layer of a deep learning network. In order to compute the label score, the fully connected layers depend on an activation function known as the Softmax classifier. The Softmax function receives at its input an array of features as a result of the learning process. At the same time, its output is the likelihood that the image represents a specific category.

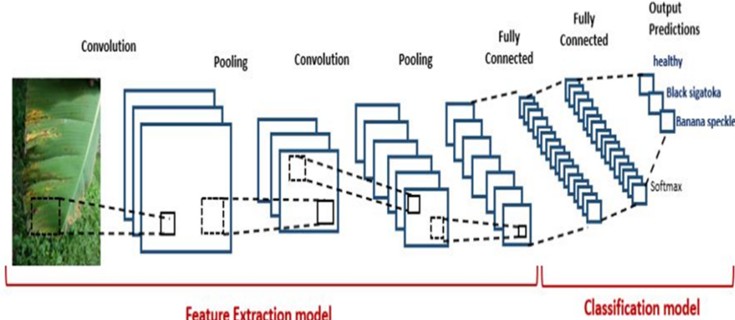

**Figure 11.** CNN Model from [34].

3.3.4. CNN Customization

The initial pre-trained layers are frozen to fine-tune the model using weights and biases previously trained on the VGG-16 dataset. For CNN customization, the last convolution and pooling layers will stay unfroze or trainable to allow the VGG-16 model to train the local dataset images. Even the original VGG-16 model aims to classify out of 1000 classes, and the proposed system has only four classes (Healthy, Multiple Diseased, Rust, and Scab).

Another pooling layer is added before the flattened layers to improve the accuracy of the computer vision system significantly. This layer finds the global average output for the extracted features in the preceding layer. The added layer will reduce the data size noticeably and prepare the model for classification at the final layer. The following are the employed TL techniques in the proposed system:

**Resenet50 algorthims:**

Consider a neural network block where we want to learn the true distribution $H$. Its input is $x$. ($x$). Let's write this as the difference (or residual):

$$R(x) = Output - Input = H(x) - x \tag{1}$$

Rearranging it we get,

$$H(x) = R(x) + x \tag{2}$$

**VGG16 algorthims:**

To calculate the error of the algorithm VGG16, we use:

$$e = \frac{1}{n} \sum_{k} minid(ci, Ck) \tag{3}$$

The ILSVRC dataset contains fixed-size 224 × 224 RGB pictures. Each image has 224 × 224 pixels values for each color channel (red, green, and blue). Each input can be represented as an x-labeled tensor with dimensions (224, 224, 3).

Each pixel's value is a scalar that can range from 0 to 225. A pixel value can be expressed as $x$ ($i$, $j$, $k$), where $I$ stands for the first dimension, $j$ for the second dimension, and $k$ for the third dimension.

The image's first two dimensions show where each pixel value is located, while the third dimension shows which channel that pixel value belongs to (so, for example, if the third dimension of a pixel value is equal to 1, it belongs to the red channel).

Thus, the following are included in this design for VGG16 algorithms:

1. Conversion Layers: In the neural network example, the change from the first layer $a[0]$ to the second layer $a[1]$ was simply accomplished by multiplying the weight matrices, including a bias, and applying the ReLU function element-by-element.

$$a[1] = g(W[1]a[0] + b[1]) \tag{4}$$

2. Pooling Layers: The pooling layer will be the subject of our next discussion. We will pay special attention to the architectural transition between the initial pooling layer ($m[2]$) and a convolution layer ($a[2]$).
   We'll break down what occurs in the pooling layer of the VGG16 architecture first and then talk about the reasons for using pooling layers. Let's begin! The pooling procedure used by VGG16 is known as "max pooling". The dimensions of the output $m[2]$ are (224, 224, 64) for the input and (224, 224, 64) for the input (112, 112, 64).
   Let's simplify the dimensions to make the example easier to follow, similar to our previous example. Let's say $m[2]$ and $a[2]$ have dimensions of 6, 6, and (3,3,6), respectively.

$$m[2]_{(i,j,k)} = max_{(i \times s <= l < i \times s + f, j \times s <= l < j \times s + f)} [2]_{(l,m,k)} \tag{5}$$

3. Softmax Layer: We utilize the softmax function to complete the transition from fully connected to the softmax layer. Next, let's talk about how the softmax function functions.
   As with most fully connected layers, the shift from fully connected layer $a[15]$ to softmax layer $a[16]$ usually begins. To obtain $z[16]$, we do matrix multiplication using $W[16]$ and add a bias $b[16]$. Given that $a[15]$ is a row vector with dimensions (4096, 1) and $z[15]$ is a row vector with dimensions (100, 1), it follows that $W[16]$ and $b[16]$ have dimensions (1000, 4096) and (100, 1). (1000, 1).

$$z[16] = W[16]a[15] + b[16] \tag{6}$$

**Incpection V3 algorthims:**

As a regularizer, Inception-v3 inserts a batch normalization (BN) layer between the auxiliary classifier and the fully connected (FC) layer. The batch gradient descent approach can be used in the BN model to quicken deep neural network training and model convergence. The BN formulas are written as follows:

$$B = \chi_{1...m}, \gamma, \beta \tag{7}$$

$$y_i = BN_{\gamma,\beta}(xi) \tag{8}$$

*x* is the minimum activation value of batch *B*, *m* is the number of activation values, and these are learnable parameters (which adjust the variance in the value distribution and adjust the position of the average value); *B* represents the average value in one dimension; 2 is the standard deviation in each dimension of the feature map; and *C* is a constant.

## 4. Results and Analysis

The accuracy scores of the Inception V3, VGG16, ResNet-50, and VGG19 deep learning algorithms were assessed. First, we used the pre-trained InceptionV3, VGG16, VGG19, and ResNet50 networks to extract feature representations from a particular layer. The SVM classifier was used to evaluate the performance of the produced in-depth features. According to the performance findings, the Inception V3 combined with the SVM model produced a maximum classification accuracy of 92.42 percent. The suggested approach achieved the most outstanding performance for identifying apple illnesses using pre-trained CNN models.

This section presents the suggested model's analysis and experimental findings. A dataset is implemented to verify the suggested model's effectiveness and compare the results with those of other studies. The model based on VGG with 16 layers (VGG-16) was implemented on an Intel® CoreTM i7 CPU@ 2.6 GHz and 16 GB RAM without employing a separate Graphics Processing Unit using the Keras Python deep learning library on top of the TensorFlow framework (GPU). Several trials were conducted to find the ideal configuration that produces the most significant outcomes.

The dataset was partitioned into training and testing data at random, with the training dataset representing 80% of the total data and the testing dataset representing the remaining 20%. The suggested architecture was put into practice using the training dataset, and the proposed structure was evaluated using the testing dataset. As previously indicated, the suggested model was used to categorize the apple leaf RGB photos using the Kaggle development environment utilizing the three most popular pre-trained models: VGG-16, ResNet-50, and Inception V3. ResNet50, Inception V3, and VGG-16 have attained accuracies of 89.96%, 92.42%, and 91.53%, respectively. The achieved accuracy of the employed pre-trained models is displayed in the next sections. VGG-16 was chosen as the accepted model because it produced a compelling result. Consequently, it was employed in additional tests where it classified the dataset into various class combinations (Healthy, Scab, Multiple-diseased, and Rust).

The 80% train-20% test split is a common convention in machine learning for splitting a dataset into training and testing sets. The reason for using this split is that it provides a good balance between having a large enough training set to learn the parameters of a model accurately and having a sufficiently large test set to evaluate the model's performance. The 80% train-20% test split is often used because it is a simple and easy way to create a train and test dataset. It provides a good balance of data for training and testing, enough data for training the model to a good level of accuracy but also enough data for testing the model's performance.

Additionally, it is common to use cross-validation techniques to get a better estimate of the performance of the model on unseen data in order to avoid overfitting, which is a common problem in machine learning. In summary, the 80% train-20% test split is a widely used and well-established machine learning convention that balances the need for a large training set and a large test set to evaluate the model's performance.

Experiment No. 1 was first carried out to evaluate the suggested model's general capacity to identify abnormalities. It performs a binary classification operation, dividing the instances into several apple leaf classifications, including four classes that are (Healthy, Scab, Multiple-diseased, and Rust). As previously indicated, each picture in the Kaggle dataset is assigned to one of the four binary classes (0001, 0010, 0100, and 1000) based on the characteristics and symptoms retrieved for each condition.

The accuracy of a model can vary depending on the specific task and dataset being used. Generally, a model's accuracy can be compared to the accuracy of other similar models proposed in the literature by using the same dataset and evaluation metrics. It is common to report the performance of a model using metrics such as accuracy, precision, recall, and F1-score. These metrics can be used to compare the performance of different models on the same dataset.

In some cases, the accuracy of a model may be lower than that obtained by other models in the literature, but it is important to consider other factors such as the complexity of the model, the amount of data used for training, and the overall performance of the model on other metrics as well. It is also important to note that many other factors can influence the performance of a model, such as the specific architecture of the model, the choice of optimizer and learning rate, the quality and size of the dataset, and the specific task being performed. Therefore, comparing the accuracy of a model to other models in the literature is not always a straightforward task and should be done with care.

In summary, The accuracy of a model can be compared with the accuracy of other similar models that have been proposed in the literature by using the same dataset and evaluation metrics, but it's important to consider other factors such as the complexity of the model, the amount of data used for training, and the overall performance of the model on other metrics.

### 4.1. Plant Disease Classification Using CNN/VGG16

The suggested model is trained using the VGG-16 network. Table 1 provides the entire trainable parameters. Figure 12 shows that the pre-trained CNN network on the ImageNet dataset produced an overall testing accuracy of 91.53 percent (i.e., 91.53 percent of the testing pictures are correctly classified), with the network producing high accuracy predictions on the majority of categories, as shown in Figure 13.

**Table 1.** VGG-16 network model parameters.

| Item | Parameters |
|---|---|
| Total parameters | 13,000,000 |
| Trainable parameters | 500,000 |
| Non-trainable parameters | 12,500,000 |

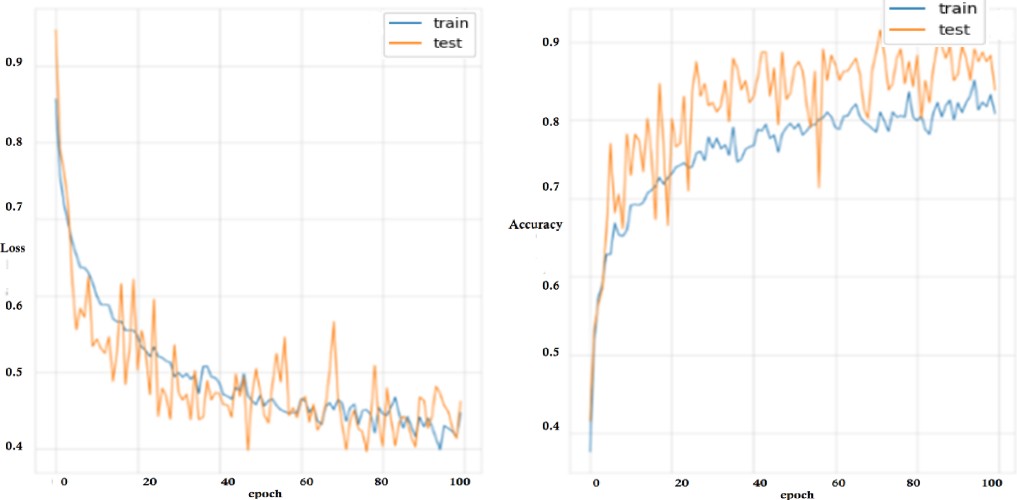

**Figure 12.** Classification Loss and Accuracy using CNN/VGG16.

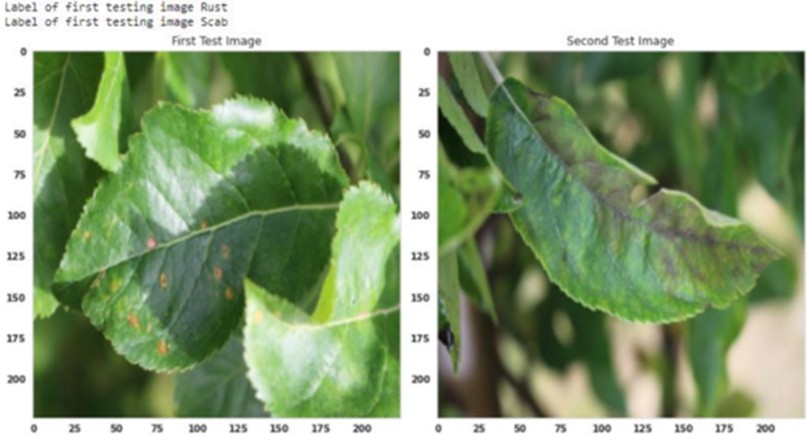

**Figure 13.** Example of the tested pictures that correctly classified.

*4.2. Plant Disease Classification Using CNN/Inception V3*

Using the Inception V3 model, we have tested the model. Table 2 lists all of the parameters and those that can be trained. As shown in Figure 14, 92.42 percent of the testing photos were properly classified based on the results of utilizing the pre-trained Inception V3 network on the ImageNet dataset.

**Table 2.** INCEPTION V3 network model parameters.

| Item | Parameters |
|---|---|
| Total parameters | 23,122,223 |
| Trainable parameters | 1,319,439 |
| Non-trainable parameters | 21,802,784 |

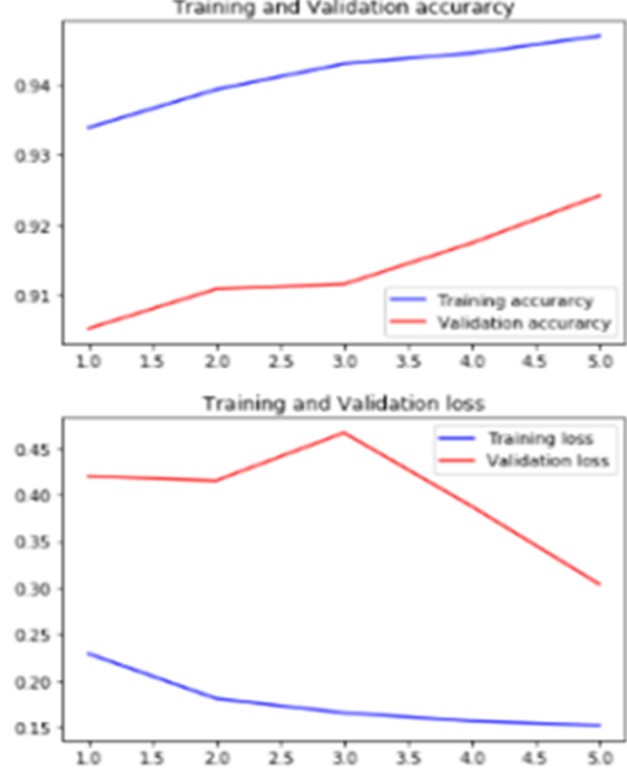

**Figure 14.** Classification Loss and Accuracy using INCEPTION V3 network model.

### 4.3. Plant Disease Classification Using CNN/ResNet50

Table 3 displays the total number of trainable parameters. The Inception V3 model has been used to test the model. According to Figure 15, the total testing accuracy attained using the pre-trained Inception V3 network on the ImageNet dataset is 89.96% (i.e., 89.96% of the testing pictures are correctly classified).

**Table 3.** ResNet network model parameters.

| Item | Parameters |
| --- | --- |
| Total parameters | 23,860,710 |
| Trainable parameters | 272,998 |
| Non-trainable parameters | 23,587,712 |

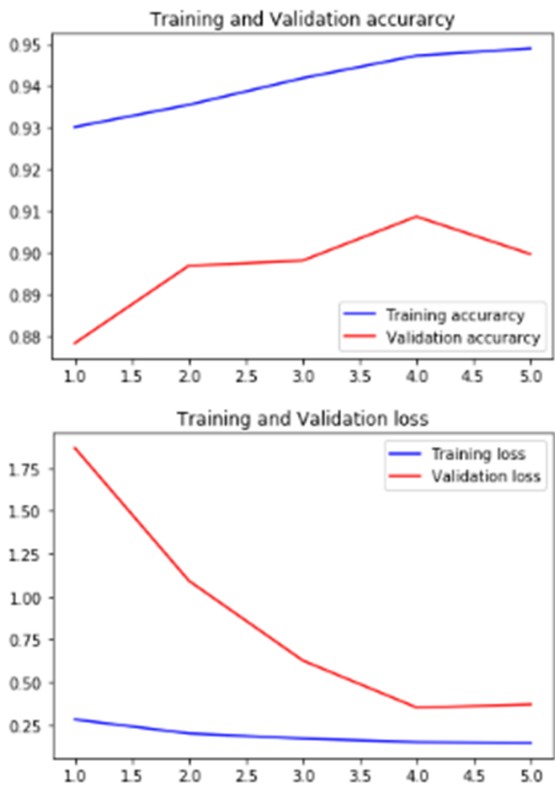

**Figure 15.** Classification Loss and Accuracy using ResNet network model.

### 4.4. Plant Disease Classification Using CNN/VGG-19

In the last model, i.e., the VGG-19 model, the total number of parameters and trainable parameters are shown in Table 4. The obtained overall testing accuracy using the pre-trained Inception V3 network on the ImageNet dataset is 90.73% (i.e., 90.73% of the testing images are correctly categorized) as illustrated in Figure 16.

**Table 4.** CNN/VGG-19 network model parameters.

| Item | Parameters |
| --- | --- |
| Total parameters | 58,102,671 |
| Trainable parameters | 58,099,791 |
| Non-trainable parameters | 2880 |

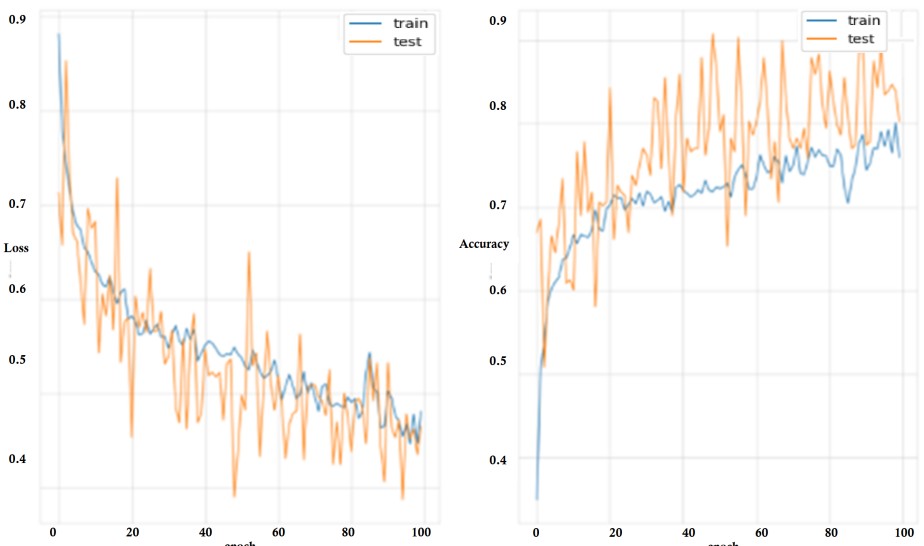

**Figure 16.** Classification Loss and Accuracy using CNN/VGG-19 network model.

Finally, it is important to note that the method described here is meant to supplement rather than replace the current methods for diagnosing apple disease. Ultimately, laboratory tests are always more trustworthy than diagnoses based solely on visual symptoms. The proposed TL models are contrasted in Table 5.

**Table 5.** A comparison between the proposed TL models.

| Model | Accuracy | Loss |
|---|---|---|
| **VGG-16** | 91.53% | 0.4785 |
| **VGG-19** | 90.73% | 0.3809 |
| **Inception V3** | 92.42% | 0.3037 |
| **ResNet** | 89.96% | 0.3693 |

*4.5. Performance Comparisons on Unseen Test Data*

The histogram-based performance analysis of applied neural network techniques presents the accuracy results of four different convolutional neural network models: VGG-16, VGG-19, Inception V3, and ResNet in Figure 17. The accuracy results indicate the percentage of correctly classified images by each model. According to the results presented in the figure, VGG-16 achieved the highest accuracy of 96%, followed by Inception V3 with an accuracy of 92%, VGG-19 with an accuracy of 90%, and ResNet with an accuracy of 89%. These results suggest that VGG-16 performed the best among the four models in classifying the images.

It's important to note that the accuracy results may vary depending on the dataset used, the preprocessing steps applied to the images, and the hyperparameters chosen for the neural network models. Therefore, the results presented in this figure should be interpreted with caution and should not be generalized to all image classification tasks. Overall, the histogram-based performance analysis of applied neural network techniques provides valuable insights into the performance of different neural network models in image classification tasks, which can be helpful in selecting the appropriate model for a particular task.

The confusion matrix-based performance comparisons of applied techniques on unseen test data are analyzed in Figure 18. The 20% testing data is utilized to build the confusion matrix analysis. The analysis validates the performance of each applied neural network technique. The confusion matrix analysis shows that the VGG-16, VGG-16, and Inception V3 models achieved lower error rates with high-performance scores. However, the ResNet technique achieved high error rates with poor performance scores for unseen

testing data. This analysis summarizes the performance of the applied neural networks for unseen testing data with target labels.

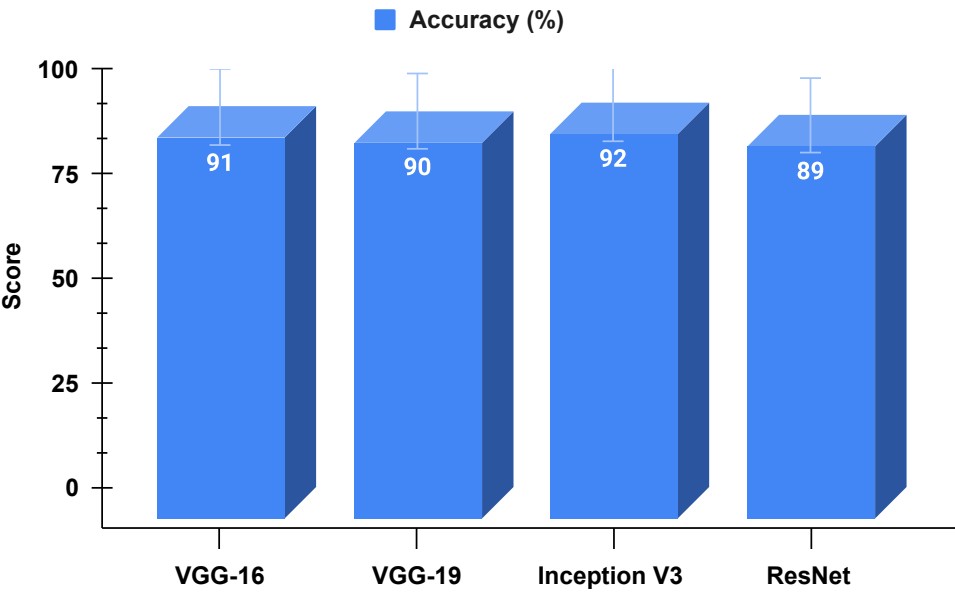

**Figure 17.** The histogram-based performance analysis of applied neural network techniques.

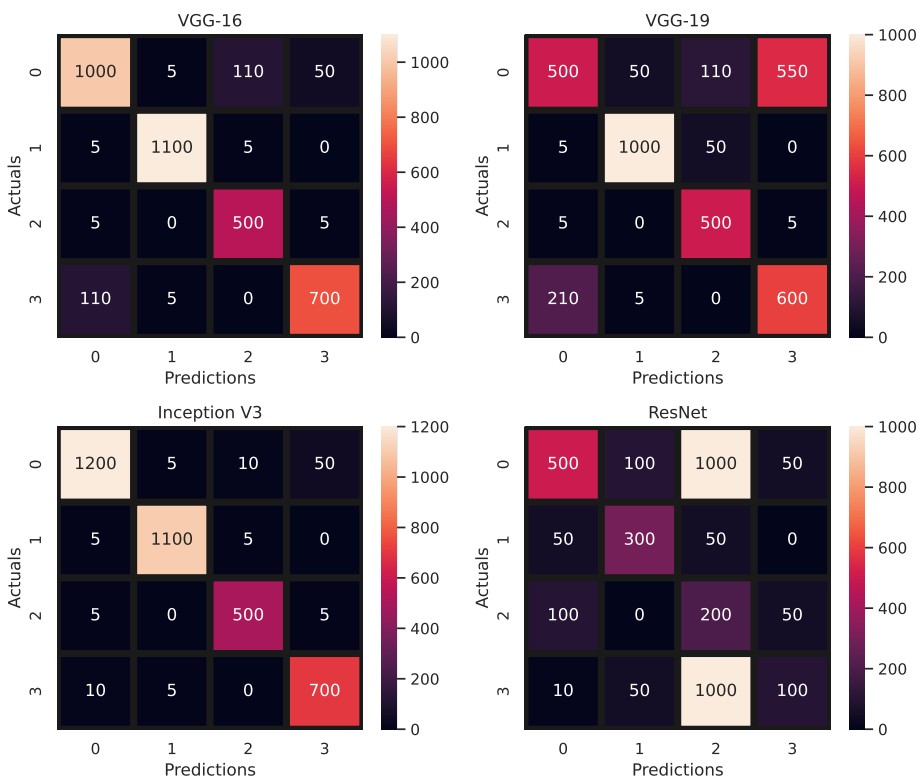

**Figure 18.** The confusion matrix analysis of applied neural network techniques.

### 4.6. Discussion

Machine learning (ML) has the potential to revolutionize the field of smart farming, particularly in the area of disease prediction for crops. The ability of ML algorithms to analyze large amounts of data and identify patterns can be used to predict the onset of disease in crops, allowing farmers to take preventative measures before the disease becomes

severe. One of the key results of using ML in smart farming for leaf disease prediction is the ability to detect diseases early, before they cause significant damage to the crop. This can greatly reduce the number of lost crops and increase overall yields. Additionally, ML-based systems can improve disease diagnosis efficiency by automating the process, reducing the time and labor required to inspect crops manually for signs of disease.

However, there are also limitations to using ML in smart farming for leaf disease prediction. One major limitation is the need for large amounts of labeled data to train the ML models. Without enough data, the models may not accurately predict disease onset. Additionally, the disease's complexity and the crop can make it difficult to develop accurate models. For farmers, data collection is one of the most important things to improve. By collecting high-quality, labeled Data, farmers can ensure that their ML-based systems are able to predict disease onset and take preventative measures accurately. Additionally, farmers should be aware of the limitations of ML-based systems and understand that they cannot replace human expertise in disease diagnosis.

Overall, while there are significant potential benefits to using ML in smart farming for leaf disease prediction, farmers need to be aware of the limitations of these systems and work to improve the data used for training. With the right data and understanding of the limitations, ML-based systems can be a valuable tool for farmers in preventing crop loss due to disease.

## 5. Conclusions and Future Work

As noted in this article, our group and other earlier researchers used AI in various applications. Here, we concentrated on using cutting-edge AI methods in the field of intelligent agriculture for sustainable cities. In this essay, plant diseases—a significant problem for agriculture—have been covered. The production and quality of crops would decline if the proposed approach were used in agriculture. This can have a disastrous effect on agriculture and the lack of diagnostic tools in many nations. Therefore, it's essential to identify plant illnesses early on by applying practical and affordable remedies.

Because of this, a method for identifying and categorizing plant diseases based on convolution neural networks was given in this research. To help farmers detect the illnesses affecting apple plants, the suggested model may be used as a decision support system (DSS). While waiting for the system to detect the disease kind, the farmer can take pictures of the apple leaf that exhibits symptoms. This paper's main contribution is the application of deep learning neural networks to the detection of three apple diseases, scab, multiple disease, and rust, in natural scenes and under challenging circumstances like lighting, complex backgrounds, and different resolution, size, viewpoint, and orientation of authentic scene images.

Following experiments, the system proved successful in producing accurate categorization outcomes. This has demonstrated that, with minimal computing effort, the suggested strategy may support an accurate diagnosis of leaf diseases. Inception V3 plus SVM, the classification model, outperformed the other three classification models in terms of performance. The accuracy of the suggested classification model for identifying apple illnesses is 92.42 percent. This study may add a larger dataset to raise the model's accuracy rating.

Using machine learning (ML) to predict plant diseases can be a powerful tool for the early detection and management of outbreaks. However, there are limitations to this approach. One of the main limitations is the need for large amounts of high-quality data to train the model. This can be difficult to obtain, especially for rare or emerging diseases. Additionally, the model's ability to generalize to new data may be limited if the training data is not diverse or representative of the entire population of plants. Another limitation is that ML models rely on patterns in the data, and these patterns may not always reflect the underlying biology or ecology of the disease. This can lead to inaccurate or unreliable predictions. Furthermore, the accuracy of predictions can be affected by choice of features

used in the model and the complexity of the model itself. Therefore, it is important to consider these limitations and validate the model's predictions with ground truth data.

ML has become an increasingly popular tool for predicting and preventing the spread of plant diseases in forests and parks. Government agencies, scholars, and park administrators can benefit from using ML to identify and address leaf disease issues. One potential use of ML in predicting leaf diseases is early detection and identification of outbreaks. By analyzing large amounts of data on leaf characteristics and disease symptoms, ML algorithms can identify patterns and trends that may indicate the presence of a disease. However, develop more effective treatment and management strategies. This can help government agencies make more informed decisions about allocating resources and addressing leaf diseases in their jurisdictions. By providing valuable insights into the patterns and trends related to leaf diseases, ML can help make more informed decisions and take effective actions to protect the health of the forests and parks under their care.

**Author Contributions:** Conceptualization: B.I. methodology: B.I. software: D.A. validation: D.A. formal analysis: D.A. investigation: D.A. resources: D.A. data curation: S.A. writing—original draft preparation: B.I., S.A., D.A., A.M., I.G. and L.A. writing—review and editing: B.I., S.A., D.A., A.M., I.G. and L.A. visualization: L.A. supervision: L.A. project administration: S.A. All authors have read and agreed to the published version of the manuscript.

**Funding:** This research received no external funding.

**Institutional Review Board Statement:** Not Applicable.

**Data Availability Statement:** All data used in this research will be available upon request.

**Acknowledgments:** Not Applicable.

**Conflicts of Interest:** The authors declare no conflict of interest.

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
